# Polymeric Nanocapsules as Nanotechnological Alternative for Drug Delivery System: Current Status, Challenges and Opportunities

**DOI:** 10.3390/nano10050847

**Published:** 2020-04-28

**Authors:** Siyuan Deng, Maria Rosa Gigliobianco, Roberta Censi, Piera Di Martino

**Affiliations:** School of Pharmacy, University of Camerino, Via S. Agostino 1, 62032 Camerino (MC), Italy; siyuan.deng@unicam.it (S.D.); maria.gigliobianco@unicam.it (M.R.G.); roberta.censi@unicam.it (R.C.)

**Keywords:** polymeric nanocapsule, drug delivery system, encapsulation, nanotechnology

## Abstract

Polymer-based nanocapsules have been widely studied as a potential drug delivery system in recent years. Nanocapsules—as one of kind nanoparticle—provide a unique nanostructure, consisting of a liquid/solid core with a polymeric shell. This is of increasing interest in drug delivery applications. In this review, nanocapsules delivery systems studied in last decade are reviewed, along with nanocapsule formulation, characterizations of physical/chemical/biologic properties and applications. Furthermore, the challenges and opportunities of nanocapsules applications are also proposed.

## 1. Introduction

Over the past century, nanotechnology has increasingly acquired a crucial role in drug delivery [1,2], diagnostic [3,4], biomedical imaging [5,6] and other medicine-related domains [7,8,9]. Employing nanoparticles as delivery system—currently a hot topic of nanotechnology in medical applications—has been widely studied and developed due to their biocompatibility, controlled- and targeted-release abilities [10,11]. Nanoparticles are generally defined as “solid colloidal particles with nano-dimension size (1–1000 nm)” [12,13]. Nevertheless, in the literature, the most common nanoparticle size referred to is between 100–500 nm, in order to avoid fast clearance upon intravenous administration, prolong circulation half-life, and at the same time, increase the probability of crossing various biologic barriers and preventing accumulation in capillaries and/or other organs [14,15]. Polymeric nanoparticles can not only modulate the pharmacokinetic properties of various active substances due to the subcellular size of nanoparticles, but also affect biocompatibility and/or biodegradability of polymers employed to produce the nanoparticles. Depending on their internal structure, polymeric nanoparticles may be further classified as nanospheres or nanocapsules [16]. As suggested by the name, a polymeric nanosphere has usually a regular sphere structure, which is composed of a solid polymeric matrix. In the other hand, a polymeric nanocapsule consists of a liquid/solid core coated with a polymeric shell, which is absent in the nanosphere [17]. In recent years, polymeric nanocapsules have attracted more interest in drug delivery applications, benefitting from their core-shell microstructure. Compared with polymeric nanospheres, the solid/oil core of nanocapsules can effectively increase drug-loading efficiency, while reduce the polymeric matrix content of nanoparticles [18]. In addition, the encapsulated payload can be isolated from tissue environment by the polymeric shell, thereby avoiding the degradation or burst release induced by pH, temperature, enzymes and other factors. Additionally, the polymeric shell can be functionalized by smart molecules able to interact with targeted biomolecules, thus enabling for targeting drug delivery [19,20,21].

Benefitting from above advantages, polymeric nanocapsules have been increased interest and applied in pharmaceutical field as drug delivery carriers. Since several specialized reviews have already discussed in-depth the nanotechnologies and polymeric materials for the formulation of nanocapsules [22,23], in the present review, we focused our attention on the nanocapsules delivery systems developed in last decade. Furthermore, the challenges and opportunities of nanocapsules applications will be heighted and discussed.

## 2. Current Status of Nanocapsules: Materials and Formulation Techniques

### 2.1. Common Materials for Preparing Polymeric Nanocapsules

#### 2.1.1. Polymeric Shell

Shell materials play a critical role in the development of polymeric nanocapsules to load, protect and release bioactive substances. The properties of polymers exert significant influence on the stability, encapsulation efficiency, release profile and biodistribution of the nanocapsule as drug delivery system. Biocompatible polymeric materials have been extensively considered as appropriate candidates for nanocapsules development. In most cases, these polymers should be biodegradable for the goal of payload-releasing and elimination of nanoparticles. However, non-biodegradable, but biocompatible polymers such as polyethylene glycol (PEG), polyvinyl alcohol (PVA) have also been widely used to contribute to the fabrication of nanoparticles. They can assist drug release via diffusion, thanks to their hydrophilicity. In addition, they could be cleared from blood via the reticuloendothelial system, eventually—despite not being degraded in smaller molecules [24,25]. To satisfy the different application requirements, various polymers have been employed for the formulation of nanocapsules shell; they can be classified into natural or synthetic polymers according to their source.

Polysaccharides—one of the most important categories of natural polymeric materials—have been broadly used as drug carriers, profiting from biocompatibility, gelation conditions and mucoadhesive properties. In general, polysaccharides are rich in deprotonated amino groups or carboxylic acid groups, resulting in cationic or anionic charges, thereby to form the polymeric shell by electrostatic attractive interactions [26,27,28].

Chitosan, as one of common natural polymers has been broadly used as drug carrier profiting from the biocompatibility, endogenous metabolizable degradable productions, gelation conditions and mucoadhesive properties [29,30]. Nanocapsules formed or covered by chitosan can process a cationic surface charge benefiting from the rich amino groups of chitosan. The cationic surface can improve the interaction between nanoparticles and the bacteria which process negative surface through electrostatic interactions. Poly (lactic -co-glycolic acid) (PLGA) nanocapsules with chitosan shell exhibited a better attachment to the *S. aureus* and *M. abscessus* compared to ones without chitosan [31]. Thereby, chitosan-based nanocapsules have been developed as drug delivery system for infectious disease [32]. However, for different applications, the strong cationic surface charge may induce nanoparticle aggregation, protein adsorption, as well as fast removal from the blood circulation [26,27]. Therefore, several anionic polymers have been used in cooperation with chitosan to act as nanocapsules shells in order to solve this problem, such as poly (acrylic acid) (PAA) [33], poly (vinyl alcohol) (PVA) [34] and also anionic polysaccharides [35,36].

Alginate is one of the anionic natural polysaccharides which has been developed as drug carrier, taking advantages of its biocompatibility, low immunogenicity and mild gelation conditions [37,38,39]. In addition to these well-known advantages, alginate is also a pH-responsive polymer, which can provide effective protection for payloads at acidic pH conditions, while achieving drug release at alkaline pH environment [40]. Thereby, nanocapsules fabricated by alginate have been developed as promising strategy for intestinal targeted drug delivery via oral administration [41].

As a biocompatible and biodegradable polyanionic polymer, dextran sulfate has been widely used in pharmaceutical field in drug delivery application [42]. Dextran sulfate is usually associated with chitosan to form multilayer nanocapsules by electrostatic integration [43]. Nanocapsules based on chitosan–dextran sulfate showed a good stability without the need of extra covalent agents [44]. Additionally, the drug release behavior can be modulated by controlling the ratio between chitosan and dextran. Chitosan–dextran nanoparticles formed with higher percentage of carboxymethyl dextran can increase the nanoparticle dissociation, thereby, increasing the gene release in serum or cytoplasm [45,46].

Moreover, common polysaccharides stated above, poly(cyclodextrin) [47], heparin [48] hyaluronan [49] and also other polysaccharides have been used to prepare nanocapsules as drug carrier for different pharmaceutic applications.

Protein-based polymers, as another kind of natural macromolecules, have been processed as polymeric shell for nanocapsules, due to their biocompatibility and tunable properties [50,51]. Albumin is a water soluble and biodegradable protein which shows critical role in the circulating system [52]. Human serum albumin has served as shell for nanocapsules. Besides controlling drug permeation rate, albumin corona can reduce the immunogenicity of nanoparticles and thus assist them to escape from the reorganization of reticuloendothelial system. Due to its biogenic properties, albumin can also serve as targeting ligand for albondin receptors which are overexpressed on endothelial cells of tumor blood vessels, providing an excellent drug targeting system [53]. Additionally, protein can be engineered into a hollow caged nanostructure by self-assemble of a defined number of subunits. These virus-like biomimetic nanocapsules can provide a very narrow size distribution and act as drug delivery system. For example, an HspG41C mutant protein-based nanocapsule formed through self-assembly of 24 individual monomeric proteins was developed as drug carrier for anti-cancer drug doxorubicin. The HspG41C nanocapsule can be easily prepared by self-assemble procedure in water environment and processed a particle size around 12 nm with narrow size distribution. It showed a good biocompatibility and successfully delivered doxorubicin to various cancer cell lines [54]. Another protein-based caged nanostructure as drug delivery system of doxorubicin was developed by Ren et al. [55]. It was fabricated through self-assembling of 60 dihydrolipolyl acyltransferase subunits (E2) and obtained a particle size around 25 nm.

Synthesized materials were indicated advantages over natural materials since they have reproducible quality and purity. In addition, they can be tuned with different chemical ionic, mechanical, solubility and degradability properties, adjusted according to the different pharmaceutical applications [56].

Aliphatic polyesters and relative copolymers are very common synthetic polymers and have been deeply studied and developed as drug delivery systems, due to their biocompatibility and biodegradability. The most common polyesters include poly (lactic acid) (PLA), poly (lactic -*co*-glycolic acid) (PLGA), poly(ε-caprolactone) (PCL). Compared with PLA and PLGA copolymers, PCLs can provide much longer degradation period. Therefore, PCLs are more suitable for long term drug delivery system or medical applications. In addition, PCLs have lower cost than PLAs and PLGAs, which is also an advantage according to some research work [19,31,57,58,59].

Additionally, Eudragit^®^ polymers are a series of synthetic polymers with different ratios of acrylate and methacrylate portions, and are used as coating materials for various kinds of dosage forms in pharmaceutical applications due to their biocompatibility, biodegradability and flexible ionic properties [56]. Eudragit^®^ polymers have been considered for the formulation of nanocapsules shell. For example, Eudragit^®^ RS100 has been used to form a polymeric shell for the delivery of dihydromyricentin. Ethyl acrylate, methyl methacrylate and a methacrylic acid esters comprise Eudragit^®^ RS100, which can provide 4.5%–6.8% quaternary ammonium groups [60]. The positive charge contributed by ammonium groups of Eudragit^®^ RS100 can neutralize the negative charge of DNA thereby crossing the cell membrane without the molecule being damaged [61].

Moreover, poly (ethylene glycol) (PEG), a biocompatible hydrophilic synthesized polymer, has been widely used as polymeric coating for nanocapsules. PEGylation can be achieved either via self-assemble nanocapsule formulation by amphiphilic PEG copolymer [62], such PLGA-PEG copolymer, or PEG coated after nanocapsules formed [63]. It has been revealed that nanocapsules with PEG exposed on the surface can improve the stability of nanocapsules in biologic media and reduce the immunogenicity [30,64,65].

#### 2.1.2. Liquid/Solid/Hollow Core

The core of nanocapsules can be hollow or consists in a liquid or solid phase, thus making it able to carry different drugs.

Nanocapsules can present an oleic core highly suitable for the encapsulation of lipophilic molecules [30]. Regarding the oily core, vegetable oil (such as soybean oil, palm oil, etc.) or fatty acids (such as medium chain triglycerides) are optimal composition oleic phase for nanocapsules fabrication, due to the capacity to dissolve lipophilic drugs and the safety of oil phase [31,66,67]. Aside from being drug dissolving media, the oil, which can simultaneously provide a therapeutic functionality, is a prompting candidate to form oleic core of nanocapsules. Copaiba oil has been used as oily core of a PCL nanocapsule in order to increase the solubility of imiquimod which is a hydrophobic anti-cancer drug. Meanwhile, copaiba oil benefits for treatments of neoplastic melanoma, micropapillary carcinoma and also has function for anti-inflammatory and analgesia [17,68]. Eudragit^®^ RL 100 nanocapsules loaded with the anti-inflammation desonide drug were developed using açai oil (AO) as oily core. More than be oil core for drug encapsulation, AO also presented anti-inflammatory and anti-proliferative effects contributing to the therapy [67]. Essential oils, such as turmeric oil and lemongrass oil, have been used as oil core due to their antibacterial, antifungal, antioxidant, antimutagenic and anticarcinogenic properties [69].

Polymeric nanocapsules can also be manufactured with aqueous core to act as promoting platform for sustained delivery of hydrophilic molecules [70]. Hydrophilic anti-cancer drugs, such as gemcitabine hydrochloride and doxorubicin, have been successfully encapsulated into polymeric nanocapsules with aqueous core [71,72]. Compared with free drug, drug-loaded nanocapsules presented higher anti-cancer effect. Additionally, the polymeric nanocapsules containing an aqueous core can also effectively encapsulate and protect the hydrophilic bioactive substance. For example, the clinical applications of mono- or oligo-nucleotides presented have been limited due to their low stability and membrane permeability in the biologic environment. The encapsulation into the aqueous core of polymeric nanocapsules can successfully protect them from the degradation in biologic fluids and promote the intracellular penetration, thereby, improve the bioavailability [73,74]. Moreover, the polymeric nanocapsules containing aqueous core can be also considered as potential delivery system for water soluble protein, such as albumin [75].

Moreover, nanocapsules can also be characterized by a hollow internal structure [76,77]. In general, nanocapsules with a hollow core are fabricated by first preparing a solid sphere that is subsequently sacrificed after polymeric shell formation. The use of solid sphere as sacrificial template can specifically provide a strong spherical framework for the assembly of nanocapsules, consisting in multiple polymer shells. Removal of the solid sphere template to provide a hollow core is necessary to modulate the in vivo drug release, and also improve the biocompatibility of the nanocapsules. The materials considered as the template core should be easily removed by mild conditions in order to avoid shell damage. Metallic materials such as iron oxide [78] and gold [33,47] spheres were used as template core and are able to be removed by hydrochloric acid solution or potassium cyanide solution after polysaccharide shell is formed. Calcium carbonate as an inorganic candidate template core of nanocapsules can be removed by adding ethylene diamine tetra acetic acid into a nanocapsule water dispersion at pH 7 [79]. In addition, silica is another inorganic material generally used as template core and removable by complete dissolution in hydrofluoric acid [80]. Additionally, polystyrene could be manufactured into nanospheres by nanoemulsion and treated with trichloromethane that is removed providing a cavity [81].

Finally, the inner core of nanocapsules can also be assembled into solid state polymeric matrix to provide loading capability for different payloads [82]. For example, nanocapsules with pectin gel core have been developed as drug delivery system for the glaucoma treatment via ocular delivery [83]. Benefiting from the porous structure and hydrophilic characteristic of pectin gel, nanocapsules exhibited efficient drug-loading yield and good patient compliance for ocular administration. However, compared with nanocapsules with oleic core or hollow core, the studies for the nanocapsules with solid core are still underdeveloped.

### 2.2. Polymeric Nanocapsule Formulation

Polymeric nanocapsule formulation techniques were exhaustively reported by Couvreur et al. [84], Mora-Huertas et al. [22] and Kothamasu et al. [23] a decade ago. Three main classical methods were described: interfacial deposition method, nanoemulsion template method and layer-by-layer method. There is no significant technical transformation during the subsequent decade and the existing three strategies have already met the most requirements of nanocapsules fabrication. The innovations in nanocapsule formulation are more focused on creative utilization of materials in order to satisfy different purposes of drug delivery applications. Various new nanocapsules created by using materials with different performances, can provide multiple loading and/or releasing capacities for encapsulated active substances. In this case, as a contribution to updating the state of knowledge without repetition, the latest developments of nanocapsules for drug delivery system in recent are summarized along with brief introduction of the classical formulation techniques.

#### 2.2.1. Interfacial Deposition Method

Interfacial deposition method, as well as nanoprecipitation, was extensively used to fabricate nanocapsules. The method was first mentioned by Fessi et al. [85] in 1989. The general nanocapsule formulation by interfacial deposition of performed polymer is described in Figure 1. Using interfacial deposition methods to prepare nanocapsules needs both organic and aqueous phase. In general, organic phase is prepared by dissolving the oil and the payload into appropriate organic solvents and subsequently adding through a thin needle into the aqueous phase. Film formed polymeric substance can be dissolved in organic or water phase according to the property of the polymer. One or more surfactants could be supplemented to increase the nanocapsules stability. Eventually, the water suspension of formed nanocapsules is obtained by removing organic solvent through diffusion or evaporation. The nanocapsules characteristics can be mainly influenced by polymer concentration, injection method of organic phase, volume ratio between organic phase and aqueous phase and also nature of materials [86,87,88]. The interfacial deposition method has been used intensively in the last decade, due to the simple operation without applied external high energy force and extensive applicability for various payloads. For example, Veragten et al. developed an olanzapine-loaded oleic core nanocapsules by interfacial deposition method which successfully presented a great mucin adhesion capacity and potential as a novel strategy for olanzapine oral administration [89]. A series of polymeric nanocapsules were developed by using various antimicrobial essential oils [90]. Interfacial deposition method can also formulate nanocapsules with a hollow core by using organic phase in absence of oleic oil but only organic solvents which can be removed by evaporation once the nanocapsules formed [82]. The studies of polymeric nanocapsule formulation by interfacial deposition methods in recent five years are summarized in Table 1.

#### 2.2.2. Nano-emulsion Template Method

As time goes on, high energy instruments were introduced in order to preform nanoemulsion thereby further fabricating into nanocapsules of different ways, usually including organic solvent diffusion/evaporation or monomers coacervation. For nanoemulsion preparation, organic or aqueous phase is emulsified in the aqueous or organic phase in presence of a surfactant, while constant energy support, such as sonication or homogenization. During the nanoemulsion formulation procedure, surfactants are driven to self-assemble at the interface between organic and inorganic phases to reduce the interfacial tension, thereby, achieving a stable state. The shell-formed polymeric materials, active substance, oil and other functional substance may be dissolved or suspended into dispersion phase or continuous phase, according to the requirement of the formulated nanocapsules [91].

##### A. Emulsion–diffusion/Evaporation Method

One of the common methods for formulating polymeric nanocapsules via nanoemulsion is the emulsion–diffusion/evaporation method. The general fabrication procedure of nanocapsules via emulsion–diffusion/evaporation method is depicted in Figure 2A. It is based on emulsification of the organic phase into an inorganic phase and subsequent elimination of the organic solvent by diffusion into the external phase or evaporation [91,92]. Nanocapsules are formed by a combination of polymer precipitation and interfacial phenomena during the diffusion—or an evaporation procedure. The polymers that can be used to formulate polymeric nanocapsules by emulsion–diffusion method must possess good solubility in an organic solvent, well miscible with water, such as acetone, ethanol or ethyl acetate, thereby, removing the organic solvent by diffusion into water [93,94]. For instance, Sombra et al. developed propionated *Sterculia striata* polysaccharide nanocapsules with a Miglyol^®^ L812 oleic core as carrier for amphotericin B [95]. The nanocapsules were formed by using acetone and methanol in organic phase, and subsequently purifying by diffusion into the water phase; the hydrophobic amphotericin B was successfully encapsulated with an excellent encapsulation efficiency of 99.2 ± 1.3%. Moreover, nonpolar solvents such as chloroform or dichloromethane—which are immiscible with water—may be used in the organic phase in the emulsion–evaporation method. A poly(DL-lactide-co-glycolide)-poly(ethylene glycol) nanocapsule was fabricated by dissolving the polymers into dichloromethane as organic phase, and then form nanoemulsion by homogenized organic phase into water phase [96]. The dichloromethane was purified by evaporation. It is also worth noting that the liquid core of this polymeric nanocapsule was composed by perfluorooctyl bromide which can act as cavitation nucleus under ultrasound exposure and used to monitor the nanocapsules collapse in vitro and in vivo. The studies of polymeric nanocapsule formulation by emulsion–diffusion/evaporation methods in recent five years are summarized in Table 2.

##### B. Emulsion–coacervation Method

Emulsion–coacervation process is also based on nanoemulsion as template for nanocapsule formulation. However, compared with emulsion–diffusion/evaporation method, the different is the polymeric shell is formed and stabilized by either physical coacervation or chemical cross-linking [97,98,99,100] (Figure 2B). The emulsion–coacervation methods are principally used for polyelectrolyte materials or monomer/polymer possessing cross-linking function groups for nanocapsules fabrication. Electrostatic interaction as a physical coacervation strategy has been used to prepare a polyelectrolyte nanocapsules for encapsulation of brinzolamide to treat glaucoma [83]. Chitosan and pectin which carries positive and negative charges, respectively, have been prepared into nanoemulsion, coacervated with each other by electrostatic attraction, and eventually formed polymeric shell of nanocapsules. Compared with physical coacervation, polymeric shell crosslinked by chemical approach may provide more stable nanocapsule system. A cross-linked starch nanocapsules with aqueous core was developed as delivery system for hydrophilic dye by interfacial polymerization carried out using water in oil (W/O) mini-emulsion method [101]. It indicates that with higher concentration of cross-linker, the nanocapsules can form a thinker polymeric shell to minimize the leakage of hydrophilic payload to the aqueous phase. Additionally, free radical polymerization, such as reversible addition fragmentation chain transfer polymerization (RAFT) or atom transfer radical polymerization (ATRP),has been applied to develop polymeric nanocapsules via emulsion–coacervation method. Typically, an amphiphilic copolymer based on same monomer further used for polymeric shell was synthesized to act as template core and also stabilizer for nanoemulsion preparation. Subsequently, the monomer was added along with cross-linker and initiator and started to form polymeric shell via free radical polymerization. Then the pre-synthesized amphiphilic copolymer was removed by hydrolysis to provide hollow inner structure [102,103,104]. For example, pH sensitive polymeric nanocapsules were formulated by using RAFT/emulsion polymerization. A random copolymer consisting of butyl acrylate and acrylic acid was pre-synthesized as macroRAFT agent [105,106]. N,N-(dimethylamino)ethyl methacrylate or tertiary butyl methacrylate with methyl methacrylate were used to preform nanocapsules. The hollow core was obtained by hydrolysis of trifluoroacetic acid. The nanocapsules showed a rapid drug release at pH 6.5 responding to the pH sensitive property of the polymer. In addition, other chemical reactions were also applied to preform polymeric nanocapsules by emulsion–coacervation method, such sonochemical reaction [66,107] or photopolymerization [108,109]. The studies of polymeric nanocapsules formulated by emulsion–coacervation methods in recent five years are summarized in Table 3.

##### C. Double Emulsion Method

Based on the principle of emulsion diffusion/evaporation and emulsion–coacervation method, resulting nanoemulsion can be continually emulsified into a third phase to perform emulsions of emulsions, as well as double emulsion method. Double emulsions are classified into two major types, water in oil in water (W/O/W) and oil in water in oil (O/W/O) according to the phase sequence. The critical principle of double emulsion methods is choosing suitable surfactants to provide good stability between the interfaces of both internal and external emulsion [110,111]. The polymeric shell can be formed by diffusion, evaporation, coacervation or combination of these approaches. A fluorescent polymeric nanocapsule was developed by Campos et al. by double emulsion–evaporation method [112]. Erdmann et al. developed a poly(alkyl cyanoacrylate)-based polymeric nanocapsule by double emulsion method [113]. The polymeric shell was formed by interfacial polymerization of *n*-butyl cyanoacrylate and propargyl cyanoacrylate. W/O/W double emulsion has been wildly applied for development of nanocapsules containing aqueous core, which is beneficial to encapsulation and sustained release of hydrophilic drugs. Additionally, nanocapsules formed by double emulsion method can provide an optimal platform for encapsulation hydrophobic and hydrophilic substances simultaneously [114]. Two anticancer drugs hydrophilic doxorubicin and hydrophobic paclitaxel were encapsulated into a poly(methacrylic acid)/polyvinyl alcohol-based nanocapsules to improve cancer therapy [115]. The hydrophilic doxorubicin was loaded in the aqueous core by dissolving in the water phase of first emulsion, while the hydrophobic paclitaxel was carried in the polymeric shell by dissolving in the organic phase. Both of the hydrophilic and hydrophobic payloads achieved excellent encapsulation efficiency around 72% and 91%, respectively. Similar result was achieved by Balan et al. that hydrophilic doxorubicin and hydrophobic magnetite were incorporated into the nanocapsules simultaneously by double emulsion methods [116]. The doxorubicin was crosslinked on the chitosan-based shell by the degradable sodium tripolyphosphate, while magnetite was solidified by solvent evaporation. The studies of polymeric nanocapsule formulation by double emulsion methods in recently ten years are summarized in Table 4.

#### 2.2.3. Layer-by-Layer Method

The layer-by-layer (LBL) method has been pointed out as a promising approach for fabrication of nanocapsules comprised of multilayers. The general procedure for the LBL method for preparation of polymeric nanocapsules is described in Figure 3. The LBL method allows the targeting and releasing properties of nanocapsules to be controlled by modulating the composition and thickness of polymeric shells [117]. The mechanism of LBL nanocapsule formulation is generally using electrostatic attraction to achieve sequential deposition of polycations and polyanions on inorganic core, followed by sacrifice of template core. Payload entrapment by sacrificed-core LBL polymeric nanocapsules, commonly, accomplished after nanocapsule formulation by diffusion for hydrophilic payload and electrostatic interaction or hydrophobic effect for hydrophobic substance. Hydrophilic anticancer drug doxorubicin hydrochloride was loaded in a chitosan–pectin-based nanocapsule prepared via LBL method. The loading procedure was simply incubated empty nanocapsules into doxorubicin hydrochloride water solution for 24 hours, resulting a encapsulation efficiency around 76% [118]. Belbekhouch et al. developed multilayer nanocapsules by electrostatic absorption of cationic poly(cyclodextrin) and anionic alginate. The nanoparticle presented a hollow core formed by sacrifice of colloidal gold nanoparticle. Hydrophobic chemicals 4-hydroxy-tamoxifen was loaded by incubated ethanol solution. The results indicated that the 4-hydroxy-tamoxifen was successfully encapsulated retaining its biologic activity [47]. Likewise, an antibiotic amoxicillin was effectively entrapped into a hollow LBL polyelectrolyte nanocapsules by suspended the formulated nanocapsules in the amoxicillin buffer acetate solution. The amoxicillin was interacted with the polymeric shell by electrostatic absorption and achieved encapsulation efficiency around 62–75% [33].

Recently, LBL method has been combined with nanoemulsion method using oil in water nanoemulsion as template core for polyelectrolytes nanocapsule formulation. Lipophilic substance can be effectively loaded by dissolving in the organic phase of nanoemulsion [44]. Positive or negative charged oleic surfactants are commonly used for nanoemulsion preparation to provide a charged interfacial for the deposition of polyelectrolyte layer [16,119]. Taking advantages of the unique multilayer-core nanostructure, the nanocapsules formulated by LBL technique can also carrier different payloads at different position simultaneously. Ledo et al. developed a multilayer polymeric nanocapsule, with the ability to co-encapsulate a chemokine and an RNAi sequence for immunotherapy. The chemokine was entrapped in the lipid core, while, RNAi sequence was covered by subsequent layers of polyarginine and hyaluronic acid [120].

By possible arrangement of the layer order, the nanocapsules formulated by LBL methods can achieve diverse drug release behavior in different pH environment. Therefore, LBL methods are beneficial to formulate nanocapsules as drug delivery system for oral administration to modulating the drug release in different part of digestive tract [121,122,123]. A pH-responsive multilayer prepared by LBL methods was developed which was composed of four polyelectrolytes: poly-L-arginine, sodium alginate, chitosan and Eudragit L100. When using chitosan as outer layer, the nanocapsules showed a desirable delay of drug release at pH 1.2, while, a promising sustained release at pH 7.4 due to the protonated and deprotonated effect of chitosan at pH 1.2 and 7.4, respectively [79]. It has been demonstrated that this pH-sensitive nanocapsule can be considered as potential drug delivery system to achieve intestine targeting via oral administration. Studies of polymeric nanocapsule formulation by LBL methods in recent five years are summarized in Table 5.

## 3. Highlighted Applications of Nanocapsules as Drug Delivery System

Nanocapsules are very widely applied as delivery systems for the protection and encapsulation of proteins, peptides, enzymes, hormones, metabolites, genes and other drugs for various biomedical applications such as anti-inflammation therapy, anti-cancer therapy and immunotherapy [115,153,154,155]. Research focused on polymeric nanocapsules for pharmaceutical applications presents a dramatic increase in the past two decades (Figure 4). Applications of nanocapsules as drug delivery and targeting systems over recent decades are summarized in Table 6.

### 3.1. Improved Bioavailability of Poorly Soluble Drugs

One of the most prompting reasons to use polymeric nanocapsules as a drug delivery system is the enhancement of permeability, as well as solubility of drugs. By modulating the interaction between cells and drug, nanocapsules can effectively improve the bioavailability of drugs, as compared to unloaded free drugs. In particular, hydrophobic drugs may be prepared in water-dispersion dosage form—assisted by encapsulation of nanohydrogels—thereby to achieving specific administration methods. For example, clarithromycin is one of the FDA-approved antibiotics against respiratory disease, such as cystic fibrosis. Aerosolized administration is an effective method for clarithromycin to reach infected organs, i.e., respiratory system. However, efficacy is limited due to poor aqueous solubility. It has been found that clarithromycin-loaded PLGA/chitosan nanocapsules are significantly more effective than the same drug in free form by aerosolized administration [31].

Additionally, since most anti-cancer drugs have hydrophobic properties, various nanocapsules have been developed as delivery systems to overcome this limitation and enhance efficacy of anti-cancer therapy. The hydrophobic cancer drug imiquimod has been loaded into the copaiba oil core of poly (ε-caprolactone) nanocapsules. After encapsulation, imiquimod performed a good dispersion in water assisted by nanocapsules for cervical cancer treatment [17].

### 3.2. Sustained Delivery

For treatments of some diseases, it is necessary to maintain long-term therapeutic levels of drug concentration. However, frequent, repeated intakes of medicines by various administration methods are not only inconvenient for patients, but also limited by the side effects of drug toxicity. The core-shell structure of polymeric nanocapsules has been investigated the potential to provide a considerable inner space for maintaining relative high dosages of drugs, as the polymeric shell can possess controlled-release profiles for sustained delivery [83,156,157,158]. For example, to limit several side effects, the common oral dosage of cilostazol is 100-mg, twice per day—which represents a disadvantage in treatment compliance. Long-term release of cilostazol for treatment of peripheral arterial disease has been achieved by loading it into nanocapsules assembled by a poly(ε-caprolactone)-poly (ethylene glycol) shell and oil core [58]. Prolonged and sustained release of the drug over a period of for 6 days has been achieved without burst effects by both mechanisms of polymeric shell degradation and diffusion. The nanocapsules provided an effective method to ensure long-term, therapeutic-sufficient concentrations, while better for patients’ convenience.

### 3.3. Targeted Delivery

The polymeric shell of nanocapsules has been indicated as an ideal subject to be covered with targeting ligands that lead nanocapsules to cancer cells, immune cells or other critical cells on contact with selective destruction [159,160]. Functionalization of nanocapsules by specific bioactive ligands has emerged as an intense therapeutic strategy for cancer therapy for achieving drug accumulation at tumor-targeted regions, thereby either enhancing therapeutic effects, or reducing side effects on healthy tissue [161,162].

Common ligands used for active targeting include antibodies, peptides, nucleic aptamers, vitamins and carbohydrates [163,164,165,166]. One potent method for ligand decoration onto nanoparticles is chemical conjugation on the polymer—prioritized, and then exposed on the surface of nanocapsules by assembling the ligand-conjugated polymer into shell. In general, bioactive ligands and polymeric materials are rich in amino, carboxyl, hydroxyl and thiol groups, which act as conjugation linkages. Shi et al. [66] formulated a *ε*-poly-*L*-lysine (PLL)-based nanocapsules as a tumor targeting delivery system by using folic acid as a ligand. Folic acid was successfully grafted onto the main chain of PLL polymers by applying the reaction between carboxyl groups of folic acid to *N*-hydroxysuccinimide group of PLL.

As another example, amino groups of AS1411 aptamers were linked onto carboxymethyl chitosan via an esterification reaction, which is utilized as a tumor-cell targeting drug delivery system [82].

### 3.4. Stabilization under Harsh Environment

Nanocapsules have been investigated to provide a shielding environment of payloads such as peptides, proteins, enzymes or other bioactive substance to maintain stability against degradation induced by pH, temperature, chemical or light during delivery or fabrication procedures.

A kind of multilayer nanocapsule—fabricated by a layer-by-layer method—has been developed and has achieved intestine release via oral administration [79]. The payload was successfully protected against degradation in the first part of digestive system via nanocapsule encapsulation. The permeability of nanocapsules were modulated by a stimuli-responsive, multilayered polymeric shell made of four polyelectrolytes, including Eudragit^®^ L100, chitosan, sodium alginate, and poly-*L*-arginine, thereby presenting a delayed release in the acidic pH environment of the gastrointestinal tract, while a fast release at the pH found in the intestines.

Additionally, nanocapsules can also provide stable fabrication procedures for light-sensitive drugs. Photosensitive drugs such as desonide and ketoprofen have been encapsulated into the oil core of Eudragit^®^ RL 100 nanocapsules, which was confirmed to enable avoiding drug photodegradation under UV radiation [67,125].

### 3.5. Toxicity Moderation

For active substances, high toxicity and serious side reactions for host systems and healthy tissue is a major limitation for therapeutic treatments—especially for cancer chemotherapy and pharmacology [167,168,169]. Nanocapsules have been used as a strategy to reduce toxicity and side effects, while maintaining the pharmacological activity associated with the drug. The payload may be adequately separated from the external environment with a polymeric shell [170,171]. For example, amphotericin B is an effective drug against fungal infection, but its use is associated with a series of side effects such as fever, cardiac arrhythmia and nefrotoxicity, due to the aggregation pattern in solution. The drug has been encapsulated in the oil core of propionate *Sterculia striata* polysaccharide nanocapsules with a high loading yield, which successful reduced hepatotoxicity through a good distribution of amphotericin B-loaded nanocapsules [95].

## 4. Challenge of Nanocapsules as Drug Delivery System

According to a fair amount of theoretical and experimental work, polymeric nanocapsules have demonstrated enormous potentials and wide spaces to be applied as drug delivery systems for various biomedical applications. Nanocapsules are being implemented to overcome limitations of conventional drug delivery approaches by achieving high drug-content loading for both hydrophilic and hydrophobic drugs, as well as targeted and sustained delivery. However, there are still several technical challenges with polymeric nanocapsules that need to be further developed before industrial application.

### 4.1. Thickness Characterization of Polymeric Shell

Compared with nanospheres, the core-shell morphologic structure is one of the particular advantages of nanocapsules as drug delivery system. The thickness of the polymer shell plays a critical role in nanocapsule formulation, active substance protection as well as release profiles [66,175]. However, there is little discussion regarding the quantitative measurement of the polymeric shell of nanocapsules in most of studies; only few examples can be here reported. The main characterization method of the thickness is observation via transmission electron microscopy (TEM) [47,148]. For example, a 10-nm polymeric shell has been qualitatively estimated by TEM from argon oil/oleoyl polyoxylglycerides/Eudragit^®^ RS nanocapsules [176].

To obtain more specific values for shell thickness, mathematical calculation has also been carried out. The shell thickness of the nanocapsules formulated based on the scarified core template can be calculated as the different value between the particle radius of the completed nanocapsules and the scarified template. The thickness of polymeric shell of polydopamine/Au hollow nanocapsules was calculated as 70 nm [81].

The surface parameter variation before and after polymer coating can be also applied for the estimation of shell thickness. It has been demonstrated that variation versus salinity of the medium—which may be attributed to the nanoparticle surface—can be calculated through the Eversole and Boardman equation [177]. The thickness of dextran’s outer layer of a Miglyol^®^ 810/dextran nanocapsules was estimated as 5 nm by calculating the salinity difference using the Eversole and Boardman equation [103].

### 4.2. Organic Solvent Free Formation

Organic solvents play a critical role in polymeric nanocapsule formulation. Organic solvents have to be strictly eliminated from the formulation by purification procedure because of their toxicity in humans. Nonetheless, traces of organic solvents that cannot be adequately removed may cause toxic effects in clinical application. Additionally, nanoparticles are usually captured by immune cells—such as macrophages or mononuclear phonotypes—which indicates that the trigger of inflammation and the toxic organic solvent is considered as one of the reason for this inflammatory situation induced by nanoparticles [178]. Despite potential toxic risks, the organic solvent removal step can also influence the stability of nanoparticles [179] Therefore, it is significant and imperative to develop polymeric nanocapsule using solvent-free methods.

There are several studies that report the formulation of polymeric nanocapsules by solvent-free approach. Two formulation strategies are possible: either by using solvent-free organic phase for nanoemulsion preparation or form polymeric shell by polymerization in aqueous environment. A solvent-free protamine nanocapsule was developed as carrier for a model lipophilic anti-inflammatory agent, cyclosporine A. The nanocapsules were prepared by a nanoemulsion method. The organic phase was prepared by mixing Maisine 35–1 and Tween^®^ 80 which acted as oleic compound and surfactant, respectively. Anionic surfactant sodium deoxycholate and polyethylene glycol-40 stearate were added to the water phase. Polyethylene glycol-40 stearate was added to prevent the aggregation of nanocapsules. The organic and inorganic phases were prepared into nanoemulsion and added to protamine aqueous solution [180] along with magnetic stirring to obtain a nanocapsule formulation. Resulted protamine nanoparticles presented a particle size around 160–180 nm with a narrow distribution (PDI = 0.2). This solvent-free method can be considered as a promising strategy for nanocapsule prepared by water-soluble polymers. Another interesting example is that of Steelandt et al. who developed polymeric nanocapsules containing an oleic core and a poly-ε-caprolactone shell [181]. Poly-ε-caprolactone was heated at the glass transition temperature then mixed with oleic compound and surfactant to act as organic phase for emulsion preparation. This method can be considered as potential method for preparation of nanocapsules based on hydrophobic crystalline or amorphous polymer. 

The polymeric nanocapsules can also be prepared by monomer polymerization in organic solvent free condition to preform polymeric shell. For example, a nanocapsule was formulated by free radical polymerization of monomers acrylamide and 2-aminoethyl methacrylate hydrochloride in the deoxygenated buffer [182]. However, the polymerization method can also lead new toxic issue by residual monomers, cross linker or other chemical compounds.

### 4.3. Aggregation and Storage

Based on fabrication procedure, nanocapsules are in general yielded into aqueous dispersion. However, it has been highlighted that nanocapsules tend to be unstable and aggregate in water suspension, thereby inducing leaking of payload. In this case, to store nanocapsules into dried form is generally preferable to improve the stability of nanocapsules and also prolong the store term [183]. Moreover, to adapt preparations of particular pharmaceutical dosage forms, such as oral tablet, it is necessary to manufacture the polymeric nanocapsules suspension into dry state for further preservation and usage [129,148,184]. Spray drying as one of the common drying method has been applied for recovery of nanocapsules into solid dry state. To enhance the stability of nanocapsules by avoiding their aggregation during the drying process, protectants such as inulin or polyvinyl alcohol were used [126,185]. But the application of spray drying method may induce thermal degradation of polymeric nanocapsules and also drug loading [186]. Lyophilization carried out in presence of a lyoprotectant is another strategy for desiccation of nanocapsules [53]. However, thin polymeric shell of nanocapsules may compromise the stability of the whole delivery system because of the low working pressure, especially when nanocapsules are characterized by an oil or a hollow core [187]. After all, lyophilization is considered an expensive technique due to the experimental conditions, time and equipment need [188]. Therefore, optimization for an appropriate desiccation procedure for the stabilization and storage of polymeric nanocapsules is most urgent needed.

### 4.4. Sterilization

Sterilization of polymeric nanocapsules is also a major challenge in the development of appropriate drug delivery system for clinical trials and commercial manufactures. Several approaches have been studied to sterilize nanoparticles for in vivo experiments. Membrane filtration is a physical method to sterilize nanoparticles by using 0.22 μm filters: this method allows for the removal of microorganisms from nanocapsules water suspension [189]. This method does not need extraneous heat, radiation or chemicals which could induce damage or degradation of polymeric nanocapsules [190,191]. However, membrane filtration presents the limitation for the nanocapsules possessing particle size higher 200 nm. In addition, it could cause decrease of product yield due to filter clogging by nanocapsules. Autoclaving is another effective technique that has been applied for polymer nanoparticles sterilization via controlled temperature and pressure [192,193]. However, aggregation, morphology deformation and also chemical degradation were observed for several polymeric nanoparticles [194,195]. Additionally, size increasing was detected for nanocapsules with an oil core Miglyol^®^-based. The size increased depended either on swelling of polymeric shell or expansion of oil core under the harsh environment of temperature and pressure [196]. Gamma irradiation can provide homogeneous sterilization to inactive the microorganisms and avoid the risk of high temperature or pressure [197,198]. However, it may cause physical or chemical degeneration of either polymeric nanocapsules or loaded drugs. As a conclusion, at the present, to maintain a contaminant-free fabrication procedure of polymeric nanocapsules is the most optimal strategy for the preparation of sterilized nanocapsules. Additionally, novel and advanced strategies for sterilization of polymeric nanocapsules are imperative.

Furthermore, more novel polymeric materials, surfactants and also chemical ingredients have been used to formulate functional polymeric nanocapsules in order to meet applications in demand which are bringing more opportunities but also challenges for the polymeric nanoparticles as drug delivery systems.

## 5. Conclusions

Research of polymeric nanoparticles has widely attracted attention recently. This minireview has reported the studies and progress of polymeric nanocapsules as drug delivery system in pharmaceutical field, in last decade. To perform the specific core-shell nanostructure, various materials were used for polymeric nanocapsule formulation via the interfacial deposition method, nano-emulsion template method or lay-by layer method. The selection of polymers and formulation methods mainly depends on the characteristics of pharmaceutical ingredient and application purpose. Polymeric nanocapsules as drug delivery systems can improve the bioavailability of payloads and achieve sustained and targeted delivery. Likewise, they can also effectively reduce the harmful effects between payload and tissue environments. By loading the drug inside polymeric nanocapsules, it can protect the drug from failure or degradation caused by biologic environment. Meanwhile, it can also reduce the side-effect induced by drug to health tissue. To date, many studies have mainly focused on the development and characterization of bioactive substance-loaded polymeric nanocapsules. However, the storage and sterilization methods of formed drug-loaded polymeric nanocapsules are needed attentions and researches. Additionally, future perspectives in polymeric nanocapsules should focus on studies of using new and most performing polymers to develop advanced delivery system, thereby extending the applications of polymeric nanocapsules in pharmaceutical fields.

## Figures and Tables

**Figure 1 nanomaterials-10-00847-f001:**
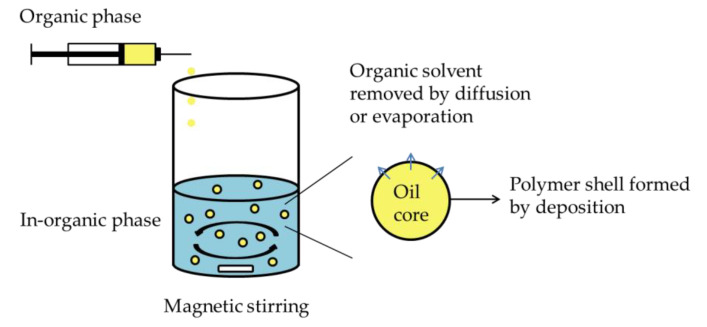
Schematic representation of nanocapsule formulation by interfacial deposition method.

**Figure 2 nanomaterials-10-00847-f002:**
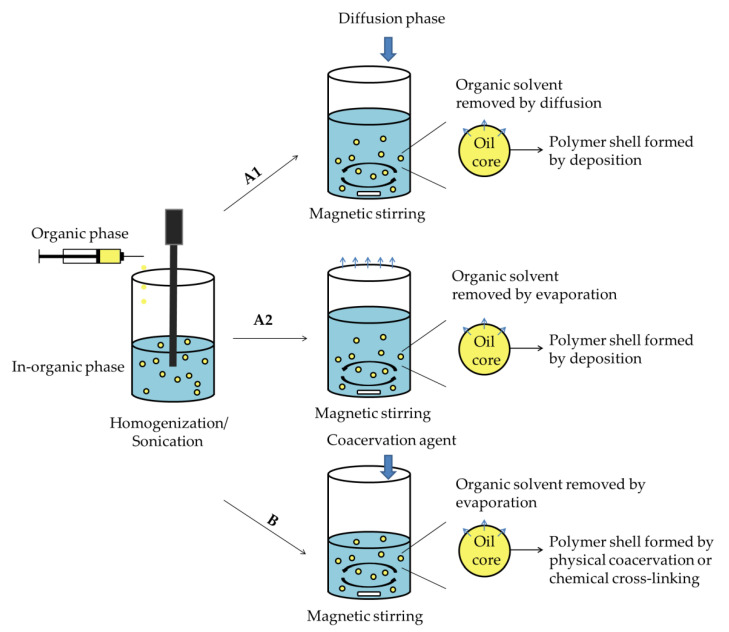
Schematic representation of nanocapsule formulation by nanoemulsion methods: (**A**). nanoemulsion–diffusion/evaporation method; (**B**). nanoemulsion–coacervation method.

**Figure 3 nanomaterials-10-00847-f003:**
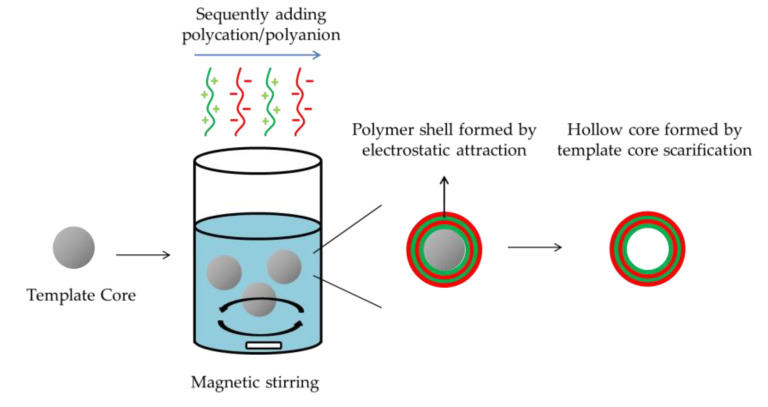
Schematic representation of nanocapsule formulation by LBL method.

**Figure 4 nanomaterials-10-00847-f004:**
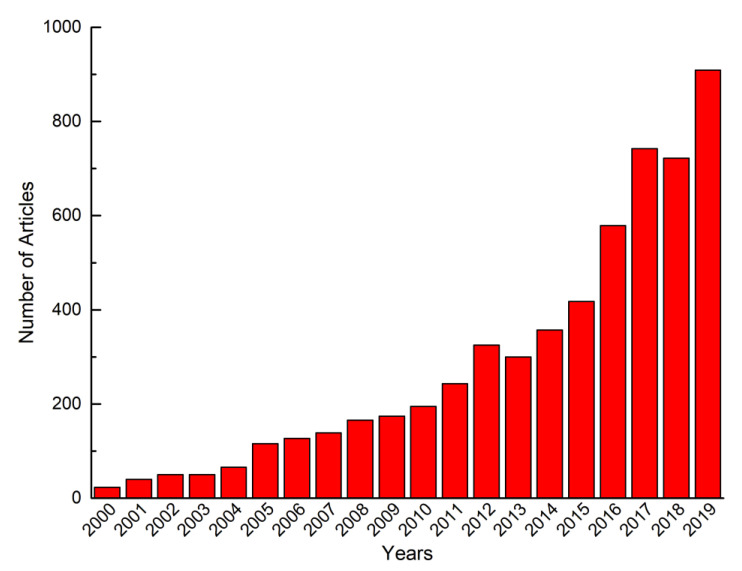
Number of articles focused on polymeric nanocapsules for medical applications (based on statistics of Science Direct).

**Table 1 nanomaterials-10-00847-t001:** Examples of polymeric nanocapsule formulation by interfacial deposition method in recent five years.

Active Substance	Polymer Film	Core	Surfactant	Organic Phase	In-organic Phase	Particle Size (nm)	PDI	Zeta Potential (mV)	Drug Loading (mg/mL)	Encapsulation Efficiency (%)	Ref.
Dutasteride	Poly-(ε-caprolactone)	Sorbitan monostearate Caprylic/capric triglyceride	Lecithin Polysorbate 80	Acetone Ethanol	Water	199.0 ± 0.5	0.12	−13.6 ± 0.6	-	≥ 96.7 ± 1.8	[124]
	Poly-(ε-caprolactone) Chitosan					224.9 ± 3.4	0.23	+40.2 ± 0.8		≥94.7 ± 3.0	
Olanzapine	Poly-(ε-caprolactone) Chitosan	Sorbitan monostearate Caprylic/capric triglycerides	Lipoid^®^ S75 Polysorbate 80	Acetone Ethanol	Water	162 ± 12	0.24 ± 0.01	+6.9 ± 0.7	1.06	42.2	[89]
Clarithromycin	Poly (lactic -*co*-glycolic acid)	Medium-chain triglycerides	Lipoid^®^ S75 Polysorbate 80	Acetone Ethanol	Water	94.9 ± 1.3	0.21 ± 0.02	−28.2 ± 0.7	0.99 ± 0.02	68 ± 1.1	[31]
	Poly (lactic -*co*-glycolic acid) Chitosan					120.6 ± 2.1	0.15 ± 0.01	+16.5 ± 0.7	0.98 ± 0.02	67 ± 0.8	
Desonide	Eudragit^®^ RL 100	Açai oil	Span^®^ 80 Polysorbate 80	Acetone	Water	165 ± 2	0.12 ± 0.03	+13.8 ± 0.3	0.26 ± 0.004	81.8 ± 1.8	[67]
		Medium chain triglyceride				131 ± 2	0.15 ± 0.02	+6.9 ± 0.7	0.26 ± 0.009	81.6 ± 0.4	
Imiquimod	Poly (ε-caprolactone)	Copaiba oil	Span^®^ 60 Tween^®^ 80	Acetone	Water	242.1 ± 17	0.17 ± 0.1	−8.9 ± 0.3	0.49 ± 0.05	98 ± 0.2	[17,68]
5-fluorouracil	Chitosan Poly(nvinylpyrrolidone- *alt*-itaconic anhydride)	Hollow	Span^®^ 80 Tween^®^ 80	Dimethyl sulfoxide Acetone	Water	118 ± 3.48	0.32 ± 0.015	−12.6 ± 0.15	0.45 ± 0.01 (g/g)	30.0 ± 0.57	[82]
	AS1411 aptamer-chitosan Poly(nvinylpyrrolidone- *alt*-itaconic anhydride)					133 ± 3.46	0.33 ± 0.020	−19.2 ± 0.12	0.34 ± 0.01 (g/g)	22.5 ± 0.86	
Ketoprofen	Eudragit^®^ S100	Rose hip oil	Span^®^ 80- Tween^®^ 80	Acetone	Water	186 ± 14	0.19 ± 0.03	−12.9 ± 2.6	1.00 ± 0.02	99.1 ± 0.5	[125]
		Medium chain triglycerides				193 ± 9	0.19 ± 0.01	−8.7 ± 1.5	0.99 ± 0.03	99.0 ± 0.2	
Cilostazol	Poly(ε-caprolactone)- Poly(ethylene glycol)	Capric/caprylic acid triglycerides	Span^®^ 80 Tween^®^ 80	Acetone	Water	130 ± 5.80	0.18 ± 0.03	−37.6 ± 1.50	11,984.64 ± 0.10 (μg/mL)	99.87	[58]
Carvedilol	Poly(ε-caprolactone)	Grape seed oil	Polysorbate 80	Acetone	Water	182 ± 7	--	−7.6 ± 1	0.47 ± 0.002	--	[126]
	Eudragit^®^ S100					139 ± 6		+7.3 ± 3	0.48 ± 0.01		
Bedaquiline	Chitosan Poly(ethylene glycol)	Oleic acid	Span^®^ 85 Tween^®^ 20	Ethanol	Water	455.6 ± 26	0.204 ± 0.020	−9 ± 2	25 ± 2 (%)	--	[30,64]
	Chitosan					328 ± 35	0.151 ± 0.027	+26 ± 4	28 ± 2 (%)	70 ± 7	
Indole-3-carbinol	Eudragit^®^ S100	Rose hip oil	Span^®^ 8 Tween^®^ 80	Acetone	Water	334 ± 1	< 2.0	+10.47 ± 0.05	0.5	44	[127]
Diphenyl diselenide	Poly(ε-caprolactone)	Medium chain triglycerides	Span^®^ 80 Tween^®^ 80	Acetone	Water	240 ± 52	0.151 ± 0.06	−10.9 ± 2.2	5	98.0	[128]
Cloxacillin benzathine	Poly(ε-caprolactone)	Labrafac^®^ CC	Span^®^80 Pluronic^®^ F68	Methanol Acetone	Water	322.4 ± 4.0	0.088 ± 0.051	−28.2 ± 0.6	0.183	46.73 ± 5.92	[129]
Antimicrobial essential oil	Cellulose acetate	Peppermint	--	Acetone	Water	~180	~0.30	~−40	82 (%)	--	[90]
		Cinnamon				~150	~0.10	~−40	14.2 (%)		
		Lemongrass				~200	~0.20	~−35	155 (%)		
p,p’-methoxyl-diphenyl diselenide	Poly(ε-caprolactone)	Medium chain triglycerides	Span^®^ 80 Tween^®^ 80	Acetone	Water	236 ± 4	0.16 ± 0.019	−5.4 ± 0.057	2.5	98.78 ± 1.54	[130,131]
Calcitriol	Poly(D,L-lactic acid)	Miglyol^®^ 829	Montanox^®^ VG80	Acetone	Water	183 ± 8	0.083 ± 0.028	−21.3 ± 2.4	--	86 ± 2	[132]
Quercetin	Chitosan	Miglyol^®^ 812	Lecithin	Ethanol	Water	190 ± 4	0.12	+ 48.4 ± 3.46	--	99 ± 1.2	[133]
Baicalein						187 ± 2	0.12	+48.1 ± 2.03		87 ± 5.1	
Meloxicam	Poly(ε-caprolactone)	Miglyol^®^ 810	Polysorbate 80	Acetone	Water	247–212	0.14	−36	99.9 (%)	--	[134]
Quinine Curcumin	Poly(ε-caprolactone)	Caprylic/capric triglyceride	Lipoid^®^ S45	Acetone	Water	194 ± 1	0.119 ± 0.00	−27.2 ± 0.1	98 ± 2.4 (%, Quinine) 96 ± 2.2 (%, Curcumin)	97 ± 2.1 (Quinine) 90 ± 1.3 (Curcumin)	[135]
Carvedilol	Poly(ε-caprolactone)	Grape oil Sorbitan monostearate	Polysorbate 80	Acetone	Water	180 ± 3	0.08 ± 0.01	−6.6 ± 0.6	--	99.1 ± 0.21	[136]
	Eudragit^®^ RS 100	Grape oil				139 ± 6	0.14 ± 0.01	+9.2 ± 2.4		88 ± 1.10	
Doxorubicin	Synthesized poly(ε-caprolactone)- poly [(methyl methacrylate)-*co*-(2-dimethylamino)ethyl methacrylate)_2_]	Caprylic triglyceride	Polysorbate 80	Acetone Ethanol	Water	60.6 ± 0.8	0.19 ± 0.01	+13.3 ± 0.9	11 μg/mL	69.7	[137]
Elisidepsin	Polyarginine	Miglyol^®^ 812 Epikuron^®^ 170	Poloxamer 188	Ethanol Acetone	Water	178 ± 15	0.1	+30 ± 11	--	46 ± 7	[138]
	Fluorescent probe DiD-labelled polyarginine					129 ± 2	0.1	+25 ± 1		75 ± 5	
	Fluorescein-DHPE-labelled polyarginine					140 ± 1	0.1	+52 ± 1		79 ± 1	
Glycerol monolaurate	Polymeric blende of poly (methyl methacrylate) and poly (ethylene glycol)	Capryc/caprylic triglyceride	Sorbitan monooleate	Acetone	Water	193.2 ± 4	0.044 ± 0.028	−23.3 ± 3	--	--	[139]
Plitidepsin	Poly-aminoacid-poly(ethylene glycol)	Epikuron^®^ 170 Miglyol^®^	Poloxamer^®^ 188	Acetone Ethanol	Water	190 ± 15	0.1	−24 ± 5	--	85 ± 4	[140]

**Table 2 nanomaterials-10-00847-t002:** Examples of polymeric nanocapsule formulation by emulsion–diffusion/evaporation method in recent five years.

Active Substance	Polymer Film	Core	Surfactant	Organic Phase	In-Organic Phase	Diffusion or Evaporation	Particle Size (nm)	PDI	Zeta Potential (mv)	Drug Loading	Encapsulation Efficiency (%)	Ref.
Amphotericin B	Propionated *Sterculia striata* polysaccharide	Miglyol^®^ L812	--	Acetone Methanol	Water	Diffusion (Water)	274.1 ± 8.6	0.181	−39.8 ± 0,2	--	99.2 ± 1.3	[95,141]
Thymoquinone Docetaxel	Chitosan	Oleic acid- phospholipid	Poloxamer	Ethanol	Water	Diffusion (Water)	141.7 ± 2.8	0.17 ± 0.02	−8.17 ± 0.1	7.7 ± 0.4 (%, Thymoquinone) 4.3 ± 0.4 (%, Docetaxel)	85.3 ± 3.1 (Thymoquinone) 66.1 ± 3.5 (Docetaxel)	[142]
Essential oil from the *L. sidoides* leaves	Poly(ε-caprolactone)	Essential oil Ethyl laurate	Kolliphor^®^ P188	Ethyl acetate	Water	Diffusion (Water)	173.6	0.2	−42	--	70.6	[143]
Docetaxel	Poly (D, L-lactide)	Labrafac^®^ CC	Polyvinyl alcohol	Ethyl acetate	Water	Diffusion (Water)	115–582	< 0.05	−36.5 ± 9	up to 68.3%	65–93	[144]
Human neutrophil elastase inhibitor	Pregelatinized modified starch	Capric/caprylic triglycerides	Tween^®^80 Cetrimide	Ethanol	water	Evaporation	100–900	--	> 30	> 0.5 (%)	> 75	[145,146]
Paclitaxel Perfluorooctyl bromide	Poly(D, L-lactide-*co*-glycolide)-poly(ethylene glycol)	Perfluorooctyl bromide	Sodium cholate	Dichloromethane	water	Evaporation	120	0.2	−18 ± 3	6 μg/mg	15 (Paclitaxel) 40 (Perfluorooctyl bromide)	[96,147]
Curcumin	Bovine serum albumin-capped gold nanoclusters	Undecylenic acid	Bovine serum albumin-capped gold nanoclusters (Self assemble)	Dichloromethane	water	Evaporation	177.54 ± 37.11	--	−29.8 ± 4.86	--	--	[13]
Catechin rich extract	Eudragit^®^ L 100	--	Sodium laureth sulfate	Methanol	Lactose aqueous solution	Evaporation	100–400	--	--	--	40–75	[94]

**Table 3 nanomaterials-10-00847-t003:** Examples of polymeric nanocapsule formulation emulsion–coacervation method in recent five years.

Active Substance	Polymer Film	Core	Surfactant	Organic Phase	In-organic Phase	Coacervation Method	Particle Size (nm)	PDI	Zeta Potential (mV)	Drug Loading (%)	Encapsulation Efficiency (%)	Ref.
Coumarin 6	Folic acid decorated-ε-poly-L-lysine-SH	Soybean oil	--	Soybean oil	Water	Sonochemical method	~500	--	--	--	--	[66]
Methotrexate	Human serum albumin Silk fibroin	n-dodecane	--	n-dodecane	Potassium phosphate buffer	Sonochemical method	438–888	--	−8.81 - −13.43	76.80 ± 17.20	98.78 ± 0.08	[107]
Brinzolamide	Chitosan Pectin	Aqueous core	Tween^®^ 80	--	Water	Electrostatic interaction	240.05 ± 0.08	0.32 ± 0.08	+27.6 ± 0.03	18.92 ± 0.25	92.20 ± 0.12	[83]
Paclitaxel	Chitosan Poly(isobutyl cyanoacrylate) (Monomer: isobutyl cyanoacrylate)	Copaiba oil	--	Ethanol	Water	Interfacial polymerization	486 ± 3	0.17	+37.1 ± 0.3	1.70 ± 0.02	74 ± 1	[148]
--	Monomer: triethylene glycol divinely ether	N-hexadecane	Sodiumdodecyl sulfate	N-hexadecane	Water	Photopolymerization	218	0.18	−71.5	--	--	[108,109]
			Polyvinyl pyrrolidone				293	0.27	−15.1			
			Pluronic^®^ PE 6100				308	0.12	−26.8			
Rhodamine B	Monomer: hydrophilic deprotonated cystamine 2,4-toluene diisocyanate	Aqueous core	Lubrizol U	Cyclohexane	Water	Interfacial Polyaddition reaction	252–444	~0.2	--	--	--	[149]
Sulforhodamine 101	Hydrophilic potato starch Cross-linker: 2,4-toluene diisocyanate	Aqueous core	Polyglycerol polyricinoleate	Cyclohexane	Sodium chloride aqueous solution	Interfacial polymerization	~200	--	--	--	~70–100	[101]
Coumarin 1	Monomer: Methyl methacrylate and 2-(diethylamino)ethyl methacrylate Initiator: 2,2-azobis (isobutyronitrile) dextran-based transurf	Miglyol^®^ 810	Amphiphilic dextran-based transurf	Methyl methacrylate	Water	RAFT polymerization	198 ± 9	0.15	an almost neutral surface	1.7	99	[103]
Bovine serum albumin	Monomer: styrene MacroRAFT agent: Polystyrene-*co*-polyN-(2-Hydroxypropyl)methacrylamide Crosslinker: divinylbenzene Initiator: 2,2’-azobis(isobutyronitrile)	Aqueous core	Polystyrene-co-polyN-(2-Hydroxypropyl)methacrylamide	Toluene *n*-hexane	Sodium chloride aqueous solution	RAFT polymerization	33–202	0.1–0.22	--	--	--	[104]
Capreomycin sulfate	Monomers: N,N-(dimethylamino)ethyl methacrylate or tertiary butyl methacrylate with methyl methacrylate MacroRAFT agent: Poly (butyl acrylate)-*co*- acrylic acid Crosslinker: Ethylene glycol dimethacrylate Initiator: azo-initiator 4,4‘ - azobis(4-cyanovaleric acid)	Aqueous core	Dimethyldioctadecyl ammonium bromide	--	Water	RAFT polymerization	~130	--	--	70	--	[105,106]
Doxorubicin	Monomer: *Tert*-butyl acrylate Macroinitiator: Folate-poly(ethylene glycol)-b-poly(tert-butyl acrylate) Reducing agent: ascorbic acid Crosslinker: N,N’-bis(acryloyl) cystamine Catalyst: CuBr_2_	Hollow	Folate-poly(ethylene glycol)-b-poly(tert-butyl acrylate)	Cyclohexanone/hexadecane	Water	AGET ATRP polymerization	~150	--	--	39.7	99.2	[102]

**Table 4 nanomaterials-10-00847-t004:** Examples of polymeric nanocapsule formulation by double emulsion method in recent ten years.

Active Substance	Polymer Film	Core	In-organic Phase 1/Surfactant	Organic Phase/Surfactant	In-organic Phase 2/Surfactant	Particle Size (nm)	PDI	Zeta Potential (mV)	Drug Loading (%)	Encapsulation Efficiency (%)	Ref.
Gemcitabine hydrochloride	Poly(ethylene glycol) Poly(L, lactide)	Aqueous core	Water/ Span	Chloroform Dichloromethane	Water/ Sodium deoxycholate hydrate	210 ± 11	0.09 ± 0.01	~−30	~22	88.6	[71]
Resiquimod (R848) Muramyl dipeptide (MDP)	Acetalated dextran	Aqueous core	Phosphate buffered saline	Dichloromethane	Phosphate buffered saline/ Polyvinyl alcohol	~150	--	~−30	0.42 (μg/mg, R848) 1.41 (μg/mg, MDP)	--	[150]
Doxorubicin hydrochloride Magnetic nanoparticles	N-palmitoyl chitosan	Aqueous core	Water/Tween^®^ 80	Chloroform	Water/Tween^®^ 80	215 ± 23.33	--	+15.8 ± 0.42	1.54	73	[116]
Doxorubicin Paclitaxel Magnetic nanoparticles	Poly(methacrylic acid) Polyvinyl alcohol	Aqueous core	Water/ Polyvinyl alcohol	Chloroform	Water/ Polyvinyl alcohol	184 ± 10	--	~10 (at pH 4–5) −15 & −30 (at pH 7)	65.21 (μg/mL, Doxorubicin) 21.74 (μg/mL Paclitaxel)	72 (Doxorubicin) 91 (Paclitaxel)	[115]
Plasmid DNA PEGylated quantum dots Pyrene Doxorubicin	Poly(styrene allyl alcohol)	Aqueous core	Water	Chloroform /Oleic acid	Water/ Polyvinyl alcohol	263 ± 42	--	--	--	~30–70 (Plasmid DNA) ~40–70 (PEGylated quantum dots) ~80–90 (Pyrene) ~10–60 (Doxorubicin)	[114]
Bovine serum albumin	Poly(lactic acid-co-glycolic acid)	Aqueous core	Water	Dichloromethane	Water/ Polyvinyl alcohol	327	--	--	--	84.75 ± 1.47	[151]
	Poly(3-hydroxybutyrateco- 3-hydroxyvalerate)					438				16.72 ± 1.06	
Fungicides tebuconazole Carbendazim	Synthesized poly(phenyleneethynylene) fluorescent polymer Poly(*ε*-caprolactone)	Hollow	Acetone (Organic solvent as first phase)	Chloroform	Water/Polyvinyl alcohol	~430	<0.2	~−13	--	--	[112]

**Table 5 nanomaterials-10-00847-t005:** Examples of polymeric nanocapsule formulation by layer-by-layer methods in recent five years.

Active Substance	Cationic Layer	Anionic Layer	Core	Phase	Particle Size (nm)	PDI	Zeta Potential (mV)	Drug Loading (%)	Encapsulation Efficiency (%)	Ref.
Curcumin	Poly-L-arginine (ARG) Chitosan (CS)	Sodium alginate (ALG) Eudragit^®^ L100 (EUD)	CaCO_3_ (Sacrificial template)	0.05 M NaCl aqueous solution	400 ± 50	~0.2	−43 ± 1.75	5.1	70	[79]
	Composition sequence: ARG-ALG-ARG-ALG-CS-EUD									
Pyrene 4-hydroxy-tamoxifen	Poly(cyclodextrin) (PCD)	Sodium alginate (ALG)	Gold (Sacrificial template)	Water	~60	--	~+20	--	--	[47]
	Composition sequence: (PCD/ALG)×4-PCD									
RNAi Chemokines CCL2	Polyarginine (ARG)	Hyaluronic acid (HA)	Glyceryl-monolete cubic gel	Water	~ 150	~0.2	−40	--	46.0 ± 1.0 (CCL2) 68.2 ± 3.2 (RNAi)	[120]
	Composition sequence: ARG-RNAi-ARG-HA									
Amoxicillin	Chitosan (CS)	Poly(acrylic acid) (PAA)	Gold (Sacrificial template)	--	<100	--	~+40 (CS out layer) ~-20 (PAA out layer)	3.5–5.5	62–75	[33]
	Composition sequence: (CS/PAA)×4 or 8									
Camptothecin	Poly-L-lysine (PLL)	Poly-L-glutamic acid sodium salt (PGA)	Hollow (Aionic surfactant docusate sodium salt)	0.015 M NaCl aqueous solution	~120	<0.2	~−5	3.2 (μg/ml)	98	[65]
	Composition sequence: (PLL/PGA)×2-PLL-PEG									
IutA antigen	Chitosan (CS)	Dextran sulfate (DEX)	Vitamin E (Oleic core)	Water	~200	~0.2	~−40	--	~70	[44]
	CS-DEX									
Indomethacin	Chitosan orPoly(ethylene imine)	Poly(acrylic acid)	Cetyltrimethylammonium bromide or hexamethylene-1,6-bis(dimethylhexadecylammoniumdibromide (Oleic core)	Water	~90–200	--	−50 – −55	~9–18	~50–60	[119]
	(PAA/PEI)×3 or 5 or 7									
Cyclosporine A	Poly-L-lysine (PLL)	Poly(L-glutamic acid) sodium salt (PGA)	Hollow (Aionic surfactant docusate sodium salt)	0.015 M NaCl aqueous solution	157.5 (PLL) 160 (PLL/PGA)	0.224 (PLL) 0.117 (PLL-PGA)	+43 −41	--	--	[152]
	PLL PLL-PGA									
Doxorubicin hydrochloride	Chitosan (CS)	Pectin (Pec)	SiO_2_ (Sacrificial template)	Water	473.10 ± 3.03	0.169	−28.33 ± 3.01	20.32 ± 0.33	76.51 ± 1.53	[118]
	(Pec/CS)×2-Pec									
Bovine serum albumin	Cationic quaternary ammonium starch (QAS)	Anionic carboxymethyl starch (CMS)	Bovine serum albumin particles	Water	20–300	--	0 – −10	--	up to 45.52	[121]
	CMS/QAS/CMS									
Clozapine	Poly-L-lysine (PLL)	Poly-L-glutamic acid (PGA)	Hollow (Aionic surfactant docusate sodium salt)	Water	~100	<0.2	−4 ± 6	--	--	[16]
	(PLL/PGA)×4-PGA-PEG									

**Table 6 nanomaterials-10-00847-t006:** Examples of polymeric nanocapsules applications as drug delivery system in recent years.

Active Substance	Therapy	Main Achieved Advantage	Polymer Membrane	Core	Formulation Method	Ref.
Olanzapine	Inhibition of startle response (PPI) in rats following oral administration	Chitosan coated increase the mucoadhesion and the brain delivery	Poly (ε-caprolactone) Chitosan	Sorbitan monostearate and caprylic/capric triglycerides	Interfacial deposition	[89]
Clarithromycin	Aerosol delivery	Improved bioavailability of poorly soluble drugs	Poly (lactic-*co*-glycolic acid)	Medium-chain triglycerides	Interfacial deposition	[31]
Desonide	Anti-inflammatory	Protecting active substance from hash environment (Photosensitive)	Eudragit^®^ RL 100	Açai oil or medium chain triglyceride	Interfacial deposition	[67]
Imiquimod	Cervical cancer	Improved bioavailability of poorly soluble drugs	Poly (ε-caprolactone)	Copaiba oil	Interfacial deposition	[17,68]
5-fluorouracil	Anti-cancer therapy	Toxicity moderation Targeting delivery (AS1411 aptamer) Improved bioavailability of poorly soluble drugs	Chitosan Poly (Nvinylpyrrolidone- *alt*-itaconic anhydride)	Hollow	Interfacial deposition	[82]
Ketoprofen	Anti-inflammatory	Protecting active substance from hash environment (Photo) Toxicity moderation	Eudragit^®^ S100	Rose hip oil	Interfacial deposition	[125]
Cilostazol	Peripheral arterial disease	Sustained delivery	Poly(ε-caprolactone)- poly (ethylene glycol)	Capric/caprylic acid triglycerides	Interfacial deposition	[58]
Carvedilol	Sublingual administration Heart failure, hypertension and coronary artery diseases	Improved bioavailability of poorly soluble drugs	Eudragit^®^ S100 Poly(ε-caprolactone)	Grape seed oil	Interfacial deposition	[126]
Bedaquiline	Mycobacterium tuberculosis	Toxicity moderation Improved bioavailability of poorly soluble drugs	Chitosan–poly (ethylene glycol)	Oleic acid	Interfacial deposition	[30,64]
Indole-3-carbinol	Antinocieptive	Protecting active substance from hash environment (Photo, temperature and pH)	Eudragit^®^ S100	Rose hip oil	Interfacial deposition	[127]
Diphenyl diselenide	Malignant cutaneous melanoma	Improved bioavailability of poorly soluble drugs Protecting active substance from hash environment (Photosensitive)	Poly(ε-caprolactone)	Medium chain triglycerides	Interfacial deposition	[128]
Cloxacillin benzathine	Mastitis	Improved bioavailability of poorly soluble drugs	Poly(ε-caprolactone)	Labrafac^®^ CC	Interfacial deposition	[129]
Curcumin	Breast cancer therapy	Improved bioavailability of poorly soluble drugs Sustained delivery	Human serum albumin	Olive oil	Interfacial deposition	[172,173]
Amphotericin B	Systemic fungal infection and leishmania	Toxicity moderation Improved bioavailability of poorly soluble drugs	Synthesized propionated *Sterculia striata* polysaccharide	Miglyol^®^ L812	Emulsion–diffusion/evaporation	[95,141]
Thymoquinone Docetaxel	Anti-cancer therapy	Toxicity moderation Improved bioavailability of poorly soluble drugs	Chitosan	Oleic acid-phospholipid	Emulsion–diffusion/evaporation	[142]
Brinzolamide	Glaucoma treatment Ocular delivery	Sustained delivery Improved bioavailability of poorly soluble drugs	Chitosan Pectin	Aqueous core	Emulsion–coacervation	[83]
Coumarin 6	Cancer	Targeting delivery (folic acid as ligand)	Folic acid decorated reductive-responsive ε-poly-L-lysine	Soybean oil	Emulsion–coacervation	[66]
Paclitaxel	Anti-cancer drug delivery via oral administration	Chitosan coated nanocapsules for oral delivery	Chitosan Poly(isobutyl cyanoacrylate)	Copaiba oil	Emulsion–coacervation	[148]
Copaiba oil	Potential mucoadhesive nanoparticles	Improved bioavailability of poorly soluble drugs	Chitosan–poly(isobutylcyanoacrylate)	Copaiba oil	Emulsion–coacervation	[174]
Curcumin	Oral administration Selective drug delivery in the colon	Protecting active substance from hash environment	Poly-L-arginine Sodium alginate Chitosan Eudragit^®^ L100	Hollow (Template CaCO_3_)	Layer-by-layer	[79]
Pyrene 4-hydroxy-tamoxifen	Anti-estrogenic delivery	Improved bioavailability of poorly soluble drugs	Poly(cyclodextrin) Sodium alginate	Hollow (Template Gold)	Layer-by-layer	[47]
RNAi Chemokines CCL2	Cancer immunotherapy	Improved bioavailability of poorly soluble drugs	Polyarginine and hyaluronic acid	Glyceryl-monooleate cubic gel	Layer-by-layer	[120]
Amoxicillin	Antibiotic therapy	Protecting active substance from hash environment (Photosensitive)	Chitosan Poly(acrylic acid)	Hollow (Template Gold)	Layer-by-layer	[33]
Camptothecin	Anti-cancer therapy	Improved bioavailability of poorly soluble drugs	Poly-L-lysine Poly-L-glutamic acid sodium salt	Hollow	Layer-by-layer	[65]
IutA antigen	—	Protecting active substance from hash environment (biochemical degradation) Improved bioavailability of poorly soluble drugs	Chitosan Dextran sulfate	Vitamin E	Layer-by-layer	[44]

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
