# Peer review of "Polymeric Nanocapsules as Nanotechnological Alternative for Drug Delivery System: Current Status, Challenges and Opportunities"

_nanomaterials, 2020, doi:10.3390/nano10050847_

Round 1

Reviewer 1 Report

The review is relatively general and superficial. Also it doesn't seem to strive for completeness.

For example many of the nanocapsule formation (not formulation) technologies require organic solvents. This is an issue, as even small amounts of residual solvents need to be removed for clinical applications. There are also completely solvent free nanocapsule formation methods (see for example the patent literature).

Several of the references contain methods leading to biocompatible but not necessarily biodegradable polymers. I believe it is not always necessary for the nanoparticles in the lower size range to be biodegradable as they will be excreted anyway. Some simple and more recent important nanocapsule formation technologies are missing in the review, for example vesicle templated polymerizations. There are even papers showing controlled drug release from those capsules (for example Loiko, O.P; van Herk, A.M.; Ali, S.I.; Burkeev, M.Z.; Tazhbayev, Y.M.; Zhaparova, L.Z. Controlled Release Of Capreomycin Sulfate From Ph-Responsive Nanocapsules, e-Polymers 2014 13 https://doi.org/10.1515/epoly-2013-0118).

Author Response

Dear Editors, Dear Reviewers,

authors replies are in the cover letter.

Best regards

Reviewer 2 Report

Dear Authors,

the minireview deals on materials and preparation methods to obtain nanocapsule for drug delivery. A summary of the principle applications was also reported and the literature of the last ten years was examinated.

Below comments point by point that authors should revise to increase the quality of the minireview.

  1. Authors should eliminate some typos.
  2. line 30. The authors should eliminate the word "pharmakodynamics" that is an intrinsic property of the drug.
  3. line 42. Authors should used a synonymous "...can be functionalized by functional..."
  4. paragraph 2.1.1, authors should introduce the use of protein before line 94, where they treated the albumin. Line 104-110, reference 52 is not appropriate. Please, should authors insert some other reference where is reported the use of PLGA as shell in addition to PCL.
  5. Line 142. Authors should added one paragraph to discuss about aqueous core nanocapsules and should reported some examples.
  6. Line 168. I suggest to introduce one other review reference"; 2002. regarding polymeric nanocapsules: Couvreur Patrick et al., "Nanocapsue Technology:a review.
  7. Authors should specify the range time of literature investigation, the generic term "recent studies" (at line 189, 206, 213, 219, 234) referred somtimes to a decade and sometimes to three years ago.

Author Response

(The authors gave the same response as above.)
